# Ramadan Fasting during Pregnancy and Health Outcomes in Offspring: A Systematic Review

**DOI:** 10.3390/nu13103450

**Published:** 2021-09-29

**Authors:** Violet N. L. Oosterwijk, Joyce M. Molenaar, Lily A. van Bilsen, Jessica C. Kiefte-de Jong

**Affiliations:** 1Leiden University College, Leiden University, 2595 DG The Hague, The Netherlands; violetoosterwijk@gmail.com (V.N.L.O.); lilyvanbilsen@hotmail.com (L.A.v.B.); 2Centre for Nutrition, Prevention and Health Services, Department of Quality of Care and Health Economics, National Institute for Public Health and the Environment, 3721 MA Bilthoven, The Netherlands; joyce.molenaar@rivm.nl; 3Department of Public Health and Primary Care, LUMC-Campus, Leiden University Medical Center, 2511 DP The Hague, The Netherlands

**Keywords:** fasting, Islam, pregnancy, humans, fetal development, pregnancy outcome, infant, newborn

## Abstract

Ramadan is one of the five pillars of Islam, during which fasting is obligatory for all healthy individuals. Although pregnant women are exempt from this Islamic law, the majority nevertheless choose to fast. This review aims to identify the effects of Ramadan fasting on the offspring of Muslim mothers, particularly on fetal growth, birth indices, cognitive effects and long-term effects. A systematic literature search was conducted until March 2020 in Web of Science, Pubmed, Cochrane Library, Embase and Google Scholar. Studies were evaluated based on a pre-defined quality score ranging from 0 (low quality) to 10 (high quality), and 43 articles were included. The study quality ranged from 2 to 9 with a mean quality score of 5.4. Only 3 studies had a high quality score (>7), of which one found a lower birth weight among fasting women. Few medium quality studies found a significant negative effect on fetal growth or birth indices. The quality of articles that investigated cognitive and long-term effects was poor. The association between Ramadan fasting and health outcomes of offspring is not supported by strong evidence. To further elucidate the effects of Ramadan fasting, larger prospective and retrospective studies with novel designs are needed.

## 1. Introduction

Ramadan is one of the five pillars of the Islam, along with pilgrimage, prayer, giving to charity and declaration of faith. During this holy month, fasting from dawn until dusk is obligatory for all healthy individuals [1]. However, the Quran exempts pregnant and lactating women from participating when abstinence from foods and fluids could pose health risks for themselves as well as their unborn babies [2]. Nevertheless, many pregnant and lactating women still engage in Ramadan fasting, due to social, cultural and religious reasons [3]. As the Ramadan follows the Lunar calendar, it always starts eleven or twelve days earlier than the year before and lasts for a totality of approximately 29 to 30 days. Practically, this implies that roughly three quarters of all children born to Muslim mothers are potentially exposed to Ramadan fasting [4].

The effect of Ramadan fasting on the health of mothers and their babies remains relatively unknown. The few studies that have investigated this topic, often report contradicting outcomes [5]. It has been suggested, for example, that Ramadan fasting influences the biophysical profile and breathing patterns of the fetus [6]. Another trend among these studies is the investigation into the effect of Ramadan fasting on health among mothers who already have pre-existing health complications such as diabetes or cancer [7,8,9,10]. However, the extent to which Ramadan fasting affects healthy, pregnant women and their offspring is less investigated. In addition, several reviews have been conducted on the effects of Ramadan fasting. However, these did not include a systematic evaluation of the available literature or merely focused on short-term perinatal outcomes without reporting on effects later in life [11,12,13]. Since both the pregnant and lactating women who wish to fast and the health care providers are relatively ignorant about the effects of Ramadan fasting, communication about decisions and advice is often poor. For example, women often choose not to inform their health care providers about their decisions to fast, out of fear for rejection and disagreement [14]. Considering the fact that the majority of infants born to Muslim mothers are exposed to the Ramadan while they are in utero, it is important to investigate whether an agreement can be established on the matter. The objective of this paper is to systematically review all literature that reported on Ramadan fasting during pregnancy and health outcomes of the offspring.

## 2. Materials and Methods

The protocol of the review was registered in PROSPERO (registration number CRD42021225935). The literature research was conducted until March 2020. Relevant literature was selected in the following databases and search engines: Web of Science, Pubmed, Cochrane Library, Embase and Google Scholar. In addition, articles were found by handsearching the reference lists of retrieved articles. There were no limits set on year of publication. In each database, the search terms included several terms for Ramadan, fasting and pregnancy and was matched with a set of terms for fetal growth indices, birth indices, cognitive effects and long-term effects (Appendix B).

### 2.1. Study Selection

All articles were screened based on their title and abstract. Relevant articles were only included when they specifically reported on health and development parameters of the offspring (i.e., fetal growth indices, birth indices, cognitive effects and long-term effects). If any uncertainty prevailed about relevance on the basis of the abstract, full-text articles were retrieved and assessed. Studies were excluded based on the following characteristics:Studies involving animals rather than humans;Studies involving women with health complications, diseases and conditions other than pregnancy;Systematic and narrative reviews;Letters, case studies, conference abstracts and editorials;Studies written in a language other than English;Qualitative analyses;Exposure to Ramadan for one day only;Studies that included other Islamic traditions other than Ramadan as exposure.

### 2.2. Data Extraction

Once all studies were identified, relevant information from each article was extracted and presented in a table (Appendix A). For each study, the design, sample size, population, location, exposure assessment, outcome assessment and findings were recorded. The following outcome measures were extracted:Fetal growth indices: estimated fetal birth weight, biparietal diameter (BPD), femur length (FL), fetal weight gain, abdominal circumference, amniotic fluid index (AFI), biophysical profile (breathing movements, gross body movements, etc.), biophysical score, non-stress Test (NST) and Doppler indicesBirth indices: birth weight, birth length, birth head circumference (HC), preterm delivery (PTD), gestational length, low birth weight (LBW), mode of delivery, fetal Apgar score, congenital anomalies, perinatal mortalityCognitive effects: test scores, mental/learning disabilities, IQ scoresLong-term effects: BMI (weight and height), diseases or conditions (diabetes, anemia, cardiovascular symptoms, breathing difficulties), general health during the life course, under-5 mortality

Due to the heterogeneity of the study designs and exposure measurements, the results could not be pooled.

### 2.3. Quality Assessment 

To rank the available articles on the basis of their quality, we used a set of criteria developed by Carter et al. [15] and the National Collaborating Centre for Methods and Tools [16]. These tools were adapted on the basis of the (ethical) feasibility of the study design (Appendix C). Each article was given zero, one or two points in five categories: study design, study size, exposure, outcome and adjustments. All articles were divided into a low score (<5), medium score (5–7) and high score (>7). In the results section, we describe the reported effects of Ramadan fasting on health outcomes of the offspring of the studies with a medium or high-quality score.

## 3. Results

### 3.1. Study Selection

Figure 1 shows the article selection process for the systematic review. The electronic database search yielded a total of 555 articles. After duplicates were removed, 396 studies remained. An additional 12 articles were added that were identified through Google Scholar and reference lists. Of these, 315 articles were excluded based on their title and abstract and another 50 were excluded after full-text assessment.

### 3.2. Study Characteristics

A total of 43 articles are categorized by outcome in Table 1. Table 1 includes all articles, however, only the medium and high-quality score studies are discussed in the results section. Nine articles investigated fetal growth indices. Twenty-seven articles focused on birth indices. Four articles explored cognitive effects. Lastly, eight articles examined the long-term effects of prenatal exposure to Ramadan. All included studies were observational, of which five were cross-sectional. There were six case-control studies identified, of which half was prospective, and twenty-one retrospective cohort studies. There were eleven prospective cohort studies. Most studies reported on birth indices. The overall quality score of the studies ranged from 2 to 9 with a mean score of 5.4.

### 3.3. Fetal Growth Indices

Figure 2 shows the results on Ramadan fasting during pregnancy and fetal growth. There was one high quality article that reported on fetal growth indices by comparing fasting and non-fasting women [23]. In this study, fasting was not found to affect biophysical profile, amniotic fluid index (AFI), the reactivity of the non-stress test (NST) and Doppler indices of the umbilical and middle cerebral arteries (MCA) (*p* > 0.05) [23]. Four medium quality studies reported on fetal growth indices and found no significant effect of fetal growth indices on most parameters [18,19,22,25]. However, three studies did suggest a significant effect of Ramadan fasting on AFI, with lower values in the fasting group compared to the non-fasting group during the second or third trimester of pregnancy [20,24,25]. In addition, Karateke et al. [20] found that the fetal biparietal diameter (BPD) increased consistently with 0.2 mm across all trimesters. Sakar et al. [24] found that among fasting women, BPD, head circumferences and femur length were significantly impaired (*p* < 0.05).

### 3.4. Birth Indices

Figure 3 illustrates an overview of the results on Ramadan fasting and birth outcomes. The studies that investigated birth indices compared different groups: fasting compared to non-fasting; fasting per trimester compared to non-fasting; trimesters without control; and fasting days and non-fasting. For accurate comparison, the groups will be discussed separately.

Concerning the studies that compared fasting to non-fasting women, there was one high quality study that found that the mean birth weight was significantly lower in the exposed offspring (108 g, *p* = 0.024) [29]. Although Kavehmanesh and Abolghasemi [34] reported a higher birth weight in the fasting group by 100 g compared to the non-fasting group (*p* = 0.009), the observed significance disappeared in the multiple regression analysis, which took into account BMI, education and age [34]. In other words, the observed significance was explained by confounding factors.

On the other hand, thirteen medium and high-quality studies [19,20,23,25,26,32,35,36,37,38,40,43,45] comparing fasting and non-fasting women found no significant differences in birth weight between the two groups. Cross et al. [31] compared Asian Muslim, white and Asian non-Muslim mothers and their offspring in Birmingham. Between these three groups, no significant difference in birth weight was found. Of these, five studies [20,32,35,37,38] found that the percentages of LBW did not differ between the fasting and non-fasting groups. In addition, half of the studies [19,29,32,34,37,38] comparing fasting vs. non-fasting women found no increased risk in preterm delivery (PTD) in either group. Two medium quality studies [42,43] compared birth weight between trimesters with a control group. After adjustment for several key confounders such as socioeconomic status (SES) and smoking status, a lower birth weight was observed in neonates who were exposed to Ramadan in their first trimester (−272.1, *p* = 0.05) [42]. However, a study conducted four years later did not report a lower birth weight between different trimesters in the fasting and control group, even though the researchers controlled for a similar set of covariates [43].

Two studies [41,47] compared the birth weight of offspring of fasting women who were in different trimesters during Ramadan without a control group. Ziaee et al. [47] found no difference in birth weight, birth length and HC between fasting and non-fasting women across different trimesters. Sarafraz et al. [41] found that birth weight was highest in the first trimester, and at its lowest in the second trimester (3411 g, *p* = 0.03). The frequency of LBW, however, was not significant between the trimesters.

Four studies [31,38,39,42] investigated the relationship between the number of days fasted and changes in birth indices. Three of these studies [31,38,41] used three comparison groups (fasted 1–10 days; 11–20 days; >20 days) and a non-fasting control group. Savitri et al. [42] compared the offspring of mothers who fasted less than half a month, more than half a month and who did not fast at all. None of these studies found a statistically significant relationship between the number of fasting days and birth indices such as (low) birth weight, mode of delivery, length and head circumference.

Five studies investigated mode of delivery [19,26,29,33,38]. One high quality study [29] and one medium quality study [26] which both investigated fasting during the second and third trimester, found that the ratio of natural delivery between the fasting and non-fasting group was significantly different, with more women having a caesarian delivery when they did not fast versus women who did. Jamilian et al. [33] and Safari et al. [38] investigated the number of fasting days and the ratio caesarean versus natural delivery, but similar to Hizli et al. [19], no statistically significant result was observed.

### 3.5. Cognitive Effects

Figure 4 shows the weight of the evidence of Ramadan fasting on cognitive effects of the offspring. A total of three medium quality studies were identified that reported on the effect of Ramadan on cognition [48,49,50]. Azizi et al. [49] did not find a significant effect on IQ scores between different groups in the case and control group in children of 4 to 13 years old. However, Majid [50] and Almond et al. [48] both found a significant effect in the first trimester, with 7.4% lower scores on cognitive tests and 8.4% lower on math tests (*p* < 0.01 for both) [41], and 0.05–0.08 standard deviations lower scores on reading, writing and mathematics respectively [48].

### 3.6. Long-Term Effects

Four medium quality studies investigated long-term effects of Ramadan fasting [51,54,57,58]. Van Ewijk [57] found that Muslims prenatally exposed to Ramadan fasting had poorer general health (6.1% of a standard deviation, *p* < 0.01), rated by professional health workers who measured a number of health and physical indicators such as height, weight, blood pressure, hemoglobin levels and lung capacity on a 9-point scale. This effect was more pronounced in those older than 45 years (18.5%, *p* < 0.01) and it appeared stronger when Ramadan started in the second trimester. Alwasel et al. [51] similarly found that boys and girls of mothers exposed to Ramadan had a significantly different length and length of gestation than their counterparts in the second trimester only (*p* = 0.005 and *p* = 0.04 respectively). Based on the same Indonesian Family Life Survey, Van Ewijk and colleagues found that on average, adult Muslims who were exposed to Ramadan in mid- and late-gestation had a lower BMI because of lower weight (respectively −0.93 kg, CI 95% −1.72, −0.14 and −1.06 kg, CI 95%–1.88, −0.25) compared to those not in utero during Ramadan [58]. Furthermore, those conceived during Ramadan, in addition to thinner stature, had a smaller stature, being on average 0.80 cm shorter than those who were not exposed. Although Kunto and Mandemakers [54] discovered a similar effect of prenatal exposure to Ramadan on BMI and stature, their results were not significant. For an overview of the evidence of long-term effects of Ramadan fasting on the offspring, see Figure 5.

## 4. Discussion

Some studies found an association between Ramadan fasting and fetal growth indices, birth indices, cognitive scores and long-term consequences. However, significant results were predominantly found in low quality studies. None of the high-quality studies reported a significant effect on fetal growth, birth, cognitive or long-term outcomes. Medium quality studies generally found mixed or non-significant results, although the majority of medium quality studies found significant results for long-term effects.

This systematic review is in line with previous reviews that concluded that Ramadan fasting did not seem to affect the health of healthy, pregnant women nor their offspring [2,11,12]. Rouhani and Azadbakht [11] did not find significant effects, however, the researchers advised to avoid Ramadan fasting due to the limitations of the studies that were included in the review. Glazier and colleagues [13] concluded that more studies are needed to accurately determine the association with maternal and neonatal outcomes. In their systematic review and meta-analysis, they found that birth weight is not adversely affected by Ramadan fasting, but there was insufficient evidence for potential effects on other perinatal outcomes. In line with Nikoo et al. [59], no reported effect of Ramadan fasting on fetal physical and mental growth was identified. However, although most studies found no association between Ramadan fasting and fetal measures, three studies found a decreased AFI [20,24,25]. Since other fetal parameters were normal, an explanation for this decrease could be maternal dehydration as a result of high temperatures and long durations of fasting. Low amniotic fluid levels have been linked to perinatal death, fetal deformations, PTD, LBW and poor neonatal health [60,61]. For this reason, it is advisable to monitor the pregnancies of these women more closely.

Neonatal anthropometry measures are considered to have predictive value for postnatal mortality [62]. Therefore, it is important to evaluate whether maternal fasting has grave implications for these outcome measures. Generally, there was no association between maternal fasting and anthropometric outcomes. In the studies that did find significant results, the association was attenuated in the multivariable analysis, the difference was very small, or there was no control for covariates that could have explained the outcome. A narrative review by Mazidi et al. [63] of 12 studies supports the conclusion that fasting has no impact on growth factors and parameters.

Van Ewijk et al. [58] was one of the first studies that established an association between in utero exposure to Ramadan fasting and thinner and shorter body sizes later in life. The developmental origins of health and disease (DOHaD) theory has been implicated in this observed relationship [64]. Maternal diet may program the baby, a phenomenon known as ‘fetal programming’ [65].

The weight of the placenta, as well as the shape of its surface, indicates the effectiveness of the placenta in transferring nutrients [65]. For this reason, placental surface size and shape at birth can be linked to chronic disease later in life [64]. A study about the famine of 1959–1961 in China after The Great Leap Forward discovered that prenatal exposure to the famine led to lower BMIs in later life [66]. A Dutch cohort study into prenatal famine showed that maternal undernutrition during pregnancy has important long-term consequences for adult health [67]. The effects depend on timing; exposure to famine in early-gestation was, among others, associated with coronary heart disease and obesity, while exposure during mid-gestation led to more microalbuminuria and obstructive airways disease. Although overall caloric intake is not dramatically reduced during Ramadan [68], it has been suggested that Ramadan fasting during pregnancy has comparable metabolic effects to those occurring during starvation or a famine [69,70]. Furthermore, previous research showed that abstaining from eating or drinking for longer than 13 h induced stress responses in mothers [71]. Thus, the Ramadan fast may also act as a stressor on the fetus. Environmental cues indicating that the external environment outside of the womb will have certain characteristics, could potentially cause genes to express a different phenotype through epigenetic mechanisms [65].

If there is a discrepancy between the in-utero environment and post-natal environment, the fetus is programmed for an adverse environment, which may not be realistic. Prenatal exposure to stress hormones may, for example, program the hypothalamic-pituitary-adrenal axis (which controls the hormonal system, including the stress response) and consequently leads to higher blood pressure and type 2 diabetes later in life [72]. This corresponds to the findings of Van Ewijk’s study, who found poorer general health, anemia and metabolic and cardiovascular risk factors in adulthood after exposure [57]. However, again, as the study was of medium quality, the results should be interpreted with caution.

For long-term, cognitive, anthropometric and health outcomes, it is important to consider that the associations may be confounded by the entire ‘Ramadan effect’ instead of the specific ‘fasting effect’. For example, during iftar (feast after sunset), the types of food consumed are often greasy, oily and sweet. Instead of the calorie restriction (fasting) causing harm to the fetus, iftar may alter the fasting effect by consumption of energy-dense foods. Furthermore, Ramadan disrupts normal nocturnal sleep patterns, particularly for those in charge of preparing the food. Clearly, the type of food consumed during iftar as well as sleeping patterns are factors that should be taken into account when discussing the ‘fasting effect’.

The search included a broad range of keyword strings that covered a wide variety of terms for Ramadan fasting, pregnancy and the effect on offspring. In addition, the search was conducted in multiple databases. However, a limitation of the review was the overall low quality of the included studies. The lack of control for key confounding factors was particularly problematic, which occurred in a large majority of the studies. Therefore, uncertainty prevailed about potential bias that could have caused any significant results. Another limitation is that multiple studies used overlap between Ramadan dates and gestation (in combination with religion, ethnicity or language) to assign exposure to Ramadan fasting. Measurements on Ramadan fasting behaviors were often not available, making it unclear if and how frequent or strict women actually fasted. This potential misclassification could have led to an under- or overestimation of the real effects of fasting [44]. Furthermore, studies that did include actual fasting in their definition of exposure to Ramadan had vastly varied measurements. For example, some studies used a certain number of fasting days, assessed (through questionnaires, religiosity) or not assessed; other studies compared women fasting for a full month to women fasting moderately or not at all. Altogether, the resulting heterogeneity impeded pooling of the overall effect.

Future research should investigate the effect of Ramadan fasting on cognition and long-term effects, ideally with a prospective study design that involves many participants. These prospective studies should be carefully designed, as the sociological and religious dimension of Ramadan may also alter maternal behavior to protect the belief that Ramadan has no effect on their offspring. Another promising approach to studying these effects would be enhanced retrospective study designs based on routine data collection comparing Islamic pregnant women during and outside the Ramadan period to reduce the effect of confounding but only under the condition that (Ramadan) fasting is registered in routine care, as other seasonal effects on birth outcomes may still apply such as malaria [73,74] and vitamin D status [75,76]. These types of studies are also needed in order to confirm the results of high quality studies in birth and fetal health outcomes. In addition to the frequency of eating during Ramadan, the quality of the food that women eat during the nighttime also deserves attention. In addition to these quantitative studies into the effects of Ramadan fasting, we argue for qualitative research into the communication between pregnant women and their healthcare providers in order to improve the shared decision-making processes concerning fasting during pregnancy.

## 5. Conclusions

In conclusion, this systematic review showed that the association between Ramadan fasting and birth and long-term outcomes is not supported by strong evidence. To further elucidate the consequences of Ramadan fasting during pregnancy, larger prospective studies and novel retrospective sibling studies taking into account socioeconomic, lifestyle and biomedical confounding factors are needed.

## Figures and Tables

**Figure 1 nutrients-13-03450-f001:**
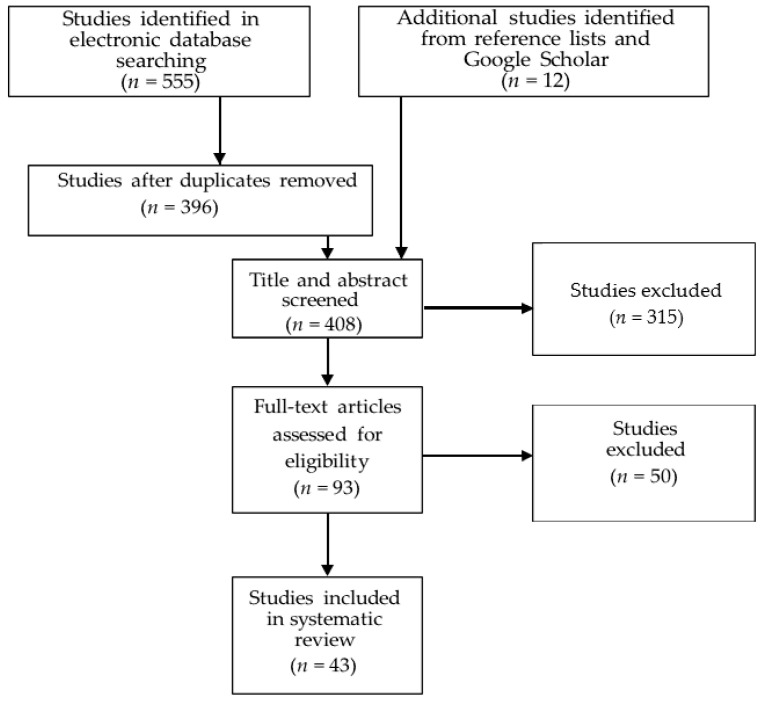
Selection process of studies for this review.

**Figure 2 nutrients-13-03450-f002:**
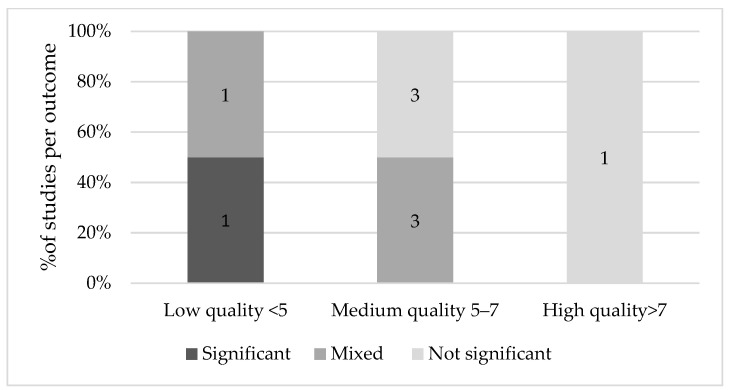
Results on Ramadan fasting during pregnancy and fetal growth indices.

**Figure 3 nutrients-13-03450-f003:**
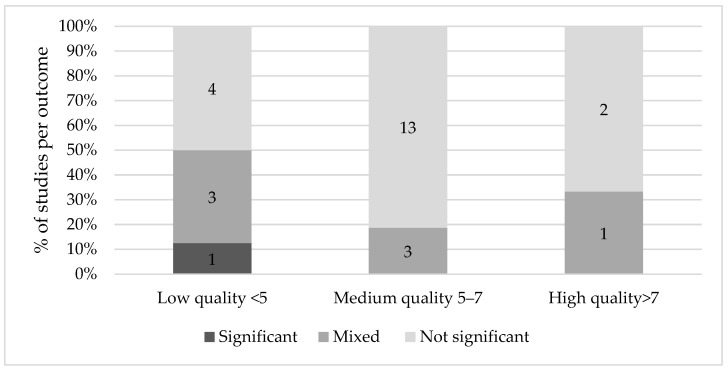
Results on Ramadan fasting during pregnancy and birth indices.

**Figure 4 nutrients-13-03450-f004:**
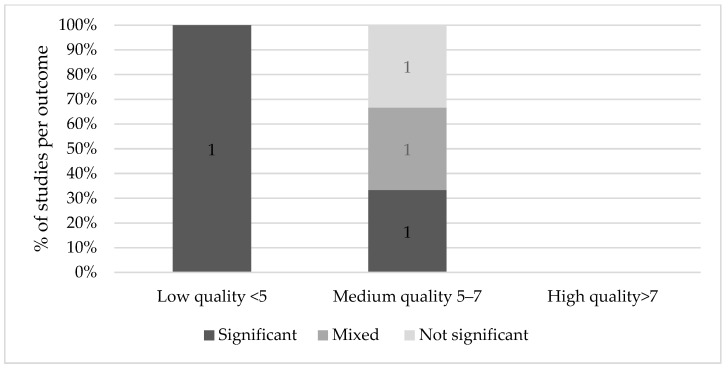
Results on Ramadan fasting during pregnancy and cognitive effects.

**Figure 5 nutrients-13-03450-f005:**
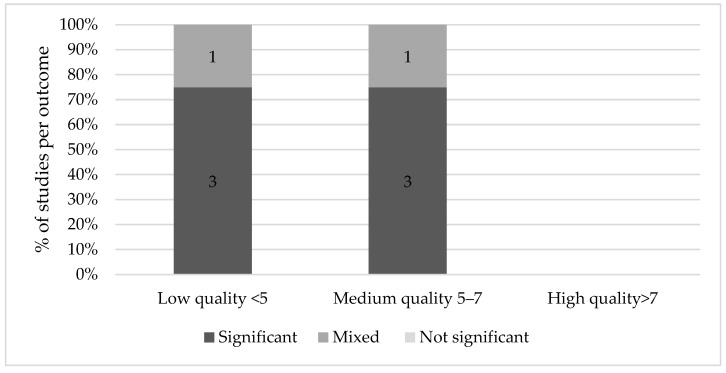
Results on Ramadan fasting during pregnancy and long-term effects.

**Table 1 nutrients-13-03450-t001:** Studies categorized based on their specific investigation.

Outcome	References
Fetal growth indices	Bayoglu Tekin et al., 2018 [17], Dikensoy et al., 2008 [18], Hizli et al., 2012 [19], Karateke et al., 2015 [20], Mirghani et al., 2003 [21], Moradi, 2011 [22], Rezk et al., 2016 [23], Sakar et al., 2015 [24], Seckin et al., 2014 [25]
Birth indices	Almond and Mazumder, 2011 [4], Altunkeser and Körez, 2016 [26], Alwasel et al., 2010 [27], Arab and Nasrollahi, 2001 [28], Awwad et al., 2012 [29], Boskabadi et al., 2014 [30], Cross et al., 1990 [31], Daley et al., 2017 [32], Hizli et al., 2012 [19], Jamilian et al., 2015 [33], Karateke et al., 2015 [20], Kavehmanesh and Abolghasemi, 2004 [34], Makvandi et al., 2013 [35], Ozturk et al., 2011 [36], Petherick et al., 2014 [37], Rezk et al., 2016 [23], Safari et al., 2019 [38], Sakar et al., 2016 [39], Sarafraz et al., 2014 [40], Sarafraz et al., 2015 [41], Savitri et al., 2014 [42], Savitri et al., 2018 [43], Savitri et al., 2019 [44], Seckin et al., 2014 [25], Shahgheibi et al., 2005 [45], Tith et al., 2019 [46], Ziaee et al., 2010 [47]
Cognitive effects	Almond and Mazumder, 2011 [4], Almond et al., 2014 [48], Azizi, 2004 [49], Majid, 2015 [50]
Long-term effects	Alwasel et al., 2011 [51], Alwasel et al., 2013 [52], Karimi and Basu, 2018 [53], Kunto and Mandemakers, 2018 [54], Pradella and van Ewijk, 2018 [55], Schoeps et al., 2018 [56], Van Ewijk, 2011 [57], Van Ewijk et al., 2013 [58]

## Data Availability

Not applicable.

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
