# Peer review of "Ramadan Fasting during Pregnancy and Health Outcomes in Offspring: A Systematic Review"

_nutrients, 2021, doi:10.3390/nu13103450_

Round 1

Reviewer 1 Report

This systematic review by Oosterwijk and colleagues appears to be a resubmission of a previous article by the authors to Nutrients (nutrients-1164474) seen previously by this reviewer.  As such, no rebuttal of the reviewers’ comments is offered.

The submission has changed very little from the previous submission, except in one major element outlined below. The minor changes are in the inclusion of the PROSPERO study registration details and restructuring of Table 1 for clarity.

A major concern that arises in comparing the present version of the manuscript with that submitted before is that the assessment of study quality has changed. For each of figures 3-6, the proportions of not significant, mixed and significant have changed in the current version compared with the previous version, often markedly. How has this come about?

According to Figure 1 in both versions of the manuscript 43 studies were included in the systematic review, so the number of papers evaluated has not changed. There has been an amendment to one of the quality statements in the ‘exposure’ domain (detailed in Appendix B): is this responsible for such a large change in quality assessment? If so, how does this retrospective change affect the interpretation of the data quality?

I am still not keen on presenting the data in terms of study quality and proportion of not significant/ mixed/significant outcomes, but if you are going to do so it is important that you indicate the number of papers falling into the low, medium and high quality categories for each outcome studied.

Reviewer 2 Report

Thank you for the opportunity to review this systematic review on the effects of Ramadan fasting on pregnancy outcomes. I was impressed by the amount of effort and care that the authors put into the synthesis of available evidence.

I only have some minor comments:

  1. Abstract: “Few medium quality studies found a significant effect on fetal growth or birth indices”. Please clarify whether the effect was positive or negative.
  2. Abstract, last line: No need to repeat the word “studies”. Keep it simple by writing “….larger prospective and retrospective studies…..”
  3. Line 170: please define “SES” which appears first time in the text
  4. The discussion is very interesting and informative. However, I think that the authors could add a paragraph highlighting the reasons behind the large heterogeneity between studies included in the analysis.

Round 2

Reviewer 1 Report

The authors have addressed my concerns.

This manuscript is a resubmission of an earlier submission. The following is a list of the peer review reports and author responses from that submission.

Round 1

Reviewer 1 Report

Oosterwijk et al. describe the outcome of a systematic review of the effects of exposure to Ramadan fasting in utero on health outcomes. The authors conclude that many of the studies conducted in this area are of poor quality, and as a result, associations between Ramadan fasting exposure and adverse health outcomes in the offspring are not supported by the published evidence. The review appears to have been conducted thoroughly and the paper is well written.

Several other reviews have been conducted in this area and similar conclusions have been drawn. Mainly that there is either no or only weak evidence that Ramadan fasting has detrimental effects on birth weight. There are insufficient studies to draw any firm conclusions on other perinatal outcomes. In this respect, Oosterwijk’s review adds nothing new to what is already known. Where they have expanded on previous reports is to consider longer term health effects and cognitive function. Unfortunately, studies in this area are limited: only 2 medium quality studies have reported on cognitive function, with contradictory outcomes, and 2 medium quality studies described reductions in general health and smaller stature.

A major weakness of the review is the absence of a concurrent meta-analysis to lend statistical weight to the conclusions that are drawn. Instead, the main findings are stratified according to the quality of the studies and are presented as not significant / mixed / significant. Other than being able to see that significant effects are generally only reported in lower quality studies, this approach does not lend itself to ready interpretation of the data. The effects of Ramadan exposure on birth weight, which is the most commonly reported variable, are summarised across studies, but without a meta-analysis it is difficult to judge the magnitude of any effect.

Why was the review protocol not registered with PROSPERO, in accordance with best practice?

In Table 1 it would be helpful to use a horizontal line or other means of dividing the references so that it is clear which are associated with a particular trait. Although they are listed alphabetically, it is not immediately obvious where the divisions fall.

Reviewer 2 Report

The authors severely misunderstand the current state of research on this topic. They review studies on Ramadan during pregnancy, classifying studies by quality using a check-list that is not appropriate here.

Comparing children of women who fasted vs did not fast during pregnancy always brings a strong risk of confounding with it. RCTs are here obviously not possible. And even prospective cohort studies (which the authors classify as those with the highest quality) need to compare two groups of women (fasting and non-fasting) who differ on a large number of characteristics. Not all of these characteristics can be measured and controlled for.

There is a large body of literature – originating among econometricians, but also including a number of studies published in high-quality epidemiological journals – that found a solution in doing a reduced-form (intent-to-treat) type of analysis, comparing children born to women who were vs were not pregnant during a Ramadan. This leads to an under estimation of effects, but avoids confounding that is related to the fact that women who CHOOSE vs DID NOT CHOOSE to fast are inherently hard to compare. These studies offer our best evidence on the effects of Ramadan during pregnancy but in the authors’ classification end up among those with the poorest quality and their results are largely dismissed by the authors. In contrast, some of the studies with a high quality ranking are actually at a high risk of residual confounding.

Not all studies mentioned in Table 1 / the supplementary material are discussed in the text at the point where one would expect them.

Reviewer 3 Report

Dear authors,

Your manuscript is well structured but I need you to answer some questions:

MATERIALS AND METHODS

  • WOS and Google Schoolar are not databases, but metasearch engines.
  • The "manual search" or reverse search is made from the reference lists of the papers that the authors have selected. What are the criteria that the authors use to say that they are relevant articles?

REFERENCES

  • Some references that have errors. The authors should review this section.